# Associations between Fatty Acid-Binding Protein 4–A Proinflammatory Adipokine and Insulin Resistance, Gestational and Type 2 Diabetes Mellitus

**DOI:** 10.3390/cells8030227

**Published:** 2019-03-08

**Authors:** Marcin Trojnar, Jolanta Patro-Małysza, Żaneta Kimber-Trojnar, Bożena Leszczyńska-Gorzelak, Jerzy Mosiewicz

**Affiliations:** 1Chair and Department of Internal Medicine, Medical University of Lublin, 20-081 Lublin, Poland; marcin.trojnar@umlub.pl (M.T.); jerzy.mosiewicz@umlub.pl (J.M.); 2Chair and Department of Obstetrics and Perinatology, Medical University of Lublin, 20-090 Lublin, Poland; jolapatro@wp.pl (J.P.-M.); b.leszczynska@umlub.pl (B.L.-G.)

**Keywords:** adipose tissue, fatty acid-binding protein 4, proinflammatory adipokine, insulin resistance, gestational diabetes mellitus, type 2 diabetes mellitus

## Abstract

There is ample scientific evidence to suggest a link between the fatty acid-binding protein 4 (FABP4) and insulin resistance, gestational (GDM), and type 2 (T2DM) diabetes mellitus. This novel proinflammatory adipokine is engaged in the regulation of lipid metabolism at the cellular level. The molecule takes part in lipid oxidation, the regulation of transcription as well as the synthesis of membranes. An involvement of FABP4 in the pathogenesis of obesity and insulin resistance seems to be mediated via FABP4-dependent peroxisome proliferator-activated receptor γ (PPARγ) inhibition. A considerable number of studies have shown that plasma concentrations of FABP4 is increased in obesity and T2DM, and that circulating FABP4 levels are correlated with certain clinical parameters, such as body mass index, insulin resistance, and dyslipidemia. Since plasma-circulating FABP4 has the potential to modulate the function of several types of cells, it appears to be of extreme interest to try to develop potential therapeutic strategies targeting the pathogenesis of metabolic diseases in this respect. In this manuscript, representing a detailed review of the literature on FABP4 and the abovementioned metabolic disorders, various mechanisms of the interaction of FABP4 with insulin signaling pathways are thoroughly discussed. Clinical aspects of insulin resistance in diabetic patients, including women diagnosed with GDM, are analyzed as well.

## 1. Introduction

Type 2 diabetes mellitus (T2DM) represents a common metabolic disorder that is characterized by chronic hyperglycemia. For more than half a century, the link between insulin resistance and T2DM has been well recognized. Insulin resistance is not only the most powerful predictor of future development of T2DM, but it is also a therapeutic target. On the other hand, gestational diabetes mellitus (GDM) is one of the most common metabolic disorders of pregnancy and its incidence has considerably increased by 10–100% in the last 20 years [1]. It should be emphasized that women with a previous history of GDM have a significantly increased risk of developing T2DM, obesity, and cardiovascular diseases in the future [2,3,4,5]. Women who had prior GDM are nearly eight times more likely to develop future T2DM compared with those with normal glucose tolerance during their pregnancy [6]. Up to one-third of women with T2DM have been previously diagnosed with GDM [7,8]. The identification of women with GDM who are at high risk of developing subsequent diseases offers a remarkable opportunity to alter their future health [1,9].

There is ample evidence to suggest a link between fatty acid-binding protein 4 (FABP4) and insulin resistance, GDM, and T2DM.

## 2. Fatty Acid-Binding Protein 4

FABP4, also referred to in the literature as adipocyte fatty acid-binding protein (AFABP), is a relatively novel adipokine [10], which belongs to the calycin protein superfamily. This protein has also been termed adipocyte P2 (aP2) since there is high sequence similarity (67%) with the myelin P2 protein (M-FABP/FABP8) [11]. FABP4 is highly expressed in adipocytes and represents approximately 1% of all soluble proteins in adipose tissue [11]. FABP4 is able to reversibly bind to hydrophobic ligands, such as saturated and unsaturated long-chain fatty acids, eicosanoids, and other lipids. Accordingly, it takes part in the regulation of lipid trafficking and responses at the cellular level [12,13,14,15]. FABPs, a family of intracellular lipid chaperones, are engaged in the transport of fatty acids to specific organelles in the cell, including mitochondria, peroxisomes, the nucleus, and the endoplasmic reticulum [12,16]. Therefore, FABPs play a significant role in lipid oxidation, lipid-mediated transcriptional regulation, and the signaling, trafficking, and synthesis of membranes. In addition, FABPs are also engaged in the regulation of the enzymatic activity and storage of lipid droplets in the cytoplasm [17], the conversion of fatty acids to eicosanoids, and the stabilization of leukotrienes [18].

The human FABP4 consists of 132 amino acids. Its molecular mass has been assessed at 14.6 kDa. FABP4 expression markedly increases at the time of adipocyte differentiation [12]. Due to the abovementioned observation, this molecule has been suggested as an adipocyte differentiation marker [19]. FABP4 expression is also enhanced during differentiation from monocytes to macrophages. A wide spectrum of different proinflammatory factors modify and control the expression of FABP4 in these cells [20]. In macrophages, FABP4 stimulates the foam cell formation. Foam cell formation, which is believed to be mediated by modified low density lipoproteins (LDLs), often occurs in the presence of increased concentrations of insulin and glucose. These increased concentrations are characteristic of the insulin resistance associated with diabetes, obesity, and the metabolic syndrome [21]. Siersbaek et al. [22] found that peroxisome proliferator-activated receptor γ (PPARγ) and CCAAT/enhancer-binding protein (C/EBP) regulate the expression of most genes associated with adipogenesis. PPARγ not only promotes the proliferation and differentiation of adipocytes, but also confers insulin sensitivity to adipocytes [23]. An increase in insulin sensitivity can in turn promote the expression of the PPARγ gene in the adipose tissue, thereby positively accelerating the differentiation of adipocytes. FABP4 expression is controlled at the transcriptional level by PPARγ and C/EBP [12]. However, the aforementioned transcriptional regulators seem to be dysregulated in T2DM [24,25].

Other actions encompass modifications of the inflammatory response mediated by activation of the IKK-NF-κB and c-Jun N-terminal kinase (JNK)-activator protein-1 (AP-1) pathways [20]. In addition, FABP4 enhances the hydrolytic activity of hormone-sensitive lipase (HSL) [26].

FABP4 as an adipokine may influence insulin sensitivity. On the other hand, the expression of FABP4 is highly induced during adipocyte differentiation and transcriptionally controlled by PPARγ agonists, fatty acids, dexamethasone, and insulin [27]. Insulin downregulates only microvesicle-free-mediated and microvesicle-secreted FABP4. However, the release of FABP4 via adipocyte-derived microvesicles is a small fraction and conveys a minor activity [27].

Furthermore, FABP4^-/-^ animals exhibit a defect in β-adrenergic stimulated insulin secretion, even under lean conditions, suggesting an effect on β cell function, another cell type that does not express FABP4. This is complemented by human studies showing that higher serum FABP4 levels correlate with a higher insulin response index in T2DM patients, and a higher insulinogenic index in non-diabetics [28]. In vivo, FABP4 levels are suppressed under refeeding or insulin [28].

Various biological effects of exogenous FABP4 have been demonstrated in different types of cells; FABP4 has been reported to enhance hepatic glucose production in vivo and in vitro and to increase the proliferation of vascular smooth muscle cells and glucose-stimulated insulin secretion in pancreatic β cells [29]. Furthermore, glucose oxidation and glycolysis are inhibited and glucose uptake as well as its utilization in the muscles and liver are considerably limited [30,31,32]. Moreover, FABP4 inhibits the activation of the insulin-signaling pathway, resulting in decreased activation of the endothelial nitric oxide synthase (eNOS) in vascular endothelial cells and nitric oxide production, inducing endothelial dysfunction [33].

## 3. Relationship between FABP4 and PPARγ

An involvement of FABP4 in the pathogenesis of obesity and insulin resistance might be mediated via FABP4-dependent PPARγ inhibition [34]. FABP4 triggers the ubiquitination and subsequent proteasomal degradation of PPARγ, a crucial regulator of adipogenesis and insulin responsiveness [35]. Importantly, FABP4-null mouse preadipocytes as well as macrophages exhibited increased expression of PPARγ, and complementation of FABP4 in the macrophages reversed the increase in PPARγ expression. The FABP4-null preadipocytes exhibited a remarkably enhanced adipogenesis compared with wild-type cells, indicating that FABP4 regulates adipogenesis by downregulating PPARγ [35].

The FABP4 level was higher and PPARγ level was lower in human visceral fat and mouse epididymal fat compared with their subcutaneous fat. Furthermore, FABP4 was higher in the adipose tissues of obese diabetic individuals compared with healthy ones. Suppression of PPARγ by FABP4 in visceral fat may explain the reported role of FABP4 in the development of obesity-related morbidities, including insulin resistance, diabetes, and atherosclerosis [35]. The most recent studies also suggest that the FABP4 inhibitor-BMS309403 significantly improves insulin sensitivity in the ob/ob mice, and the FABP4 inhibitor reduces plasma triacylglycerol levels [10].

PPARγ is activated by natural or synthetic agonists, such as the antidiabetic, thiazolidinedione (TZD) [36]. The disruption of PPARγ specifically in myeloid cells also predisposes mice to the development of diet-induced obesity, insulin resistance, and glucose intolerance, whereas activation of PPARγ within macrophages promotes lipid efflux, thereby stabilizing atherosclerotic lesions [37,38].

## 4. Relationship between FABP4 and Diseases of Civilization

It has been reported that increased circulating FABP4 levels are associated with obesity, insulin resistance, T2DM, cardiovascular disorders, arterial hypertension, cardiac dysfunction, kidney damage, fatty liver disease, and atherosclerosis [10,29,31,32,39,40,41,42]. Increased FABP4 concentrations were found in obese subjects and were positively correlated with waist circumference, blood pressure, and insulin resistance [40]. A 10-year prospective study also documented that high FABP4 levels independently predicted the development of T2DM [31]. An increased concentration of FABP4 was an independent biomarker of the development of metabolic syndrome in a five-year perspective in a Chinese population [41]. Also, Stejskal et al. [43] found that serum FABP4 levels might be a significant predictor of metabolic syndrome in a Caucasian population. Cabré et al. [44] reported that higher FABP4 plasma concentrations were associated with the early presence of metabolic syndrome components, along with inflammation and oxidation markers in T2D subjects.

Circulating FABP4 is not only a potent biomarker, but, as an adipokine, it also plays an important role in the development of metabolic syndrome and cardiovascular diseases [29]. Furthermore, FABP4 could be a treatment target in T2DM [45]. A small molecular ligand for FABP4 that blocks the binding of endogenous ligands may be developed into a drug for the treatment of T2DM [46].

## 5. Associations of FABP4 with Adipogenesis and Inflammation

FABP4 plays a crucial role in the regulation of lipid-mediated actions, such as the initialization of inflammation and oxidative stress processes [33]. The expression of this adipogenic protein can be induced by vascular endothelial growth factor (VEGF) signaling [47]. FABP4 increases the ciatriphosphate-binding cassette A1 pathway [32]. Emerging data suggests an essential involvement of FABP4 in endothelial dysfunction [47]. Exogenous FABP4 interferes with insulin stimulated production of nitric oxide in endothelial cells [48]. Cabré et al. [44] observed that increased FABP4 concentrations were associated with excessive oxidative stress and inflammatory markers in diabetes.

FABP4 negatively regulates PPARγ in macrophages and adipocytes, affecting adipocyte differentiation. Higher levels of FABP4 and lower levels of PPARγ in visceral adipose tissue, when compared with subcutaneous adipose tissue, suggest a causative link between FABP4 and the metabolic syndrome. It may also explain certain morphological and functional differences between the adipocytes found in these two types of fat tissue. Visceral preadipocytes are known to proliferate and differentiate into mature adipocytes less actively than those from subcutaneous fat. Visceral adipose tissue secretes more proinflammatory cytokines [35].

It has also been postulated that the transcription factor forkhead box protein O1 (FOXO1) is involved in lipid metabolism. The available scientific evidence suggests that metformin may have a protective effect against lipid accumulation in macrophages as it decreases FABP4 expression at the mRNA level. The exact mechanism is not yet fully understood; it may deal with decreases in transcription or by promotion of mRNA degradation. For this reason, some authors suggest that metformin should also be perceived as a therapeutic agent for the prevention and targeting of atherosclerosis in metabolic syndrome [49].

FABP4 contributes to the accumulation of short-chain free fatty acids and suppresses the activity of relevant proteins in the phosphatidylinositol 3′-kinase (PI3K)-AKT signal pathway. Accordingly, FABP4 inhibits the presence of glucose oxidation and glycolysis and decreases the uptake and utilization of glucose in human organs, such as in muscles and the liver [50]. It is known that FABP4 can bind various intracellular fatty acids and probably mediates intracellular lipid trafficking between cellular compartments [51]. It might also modulate the availability and composition of fatty acids in muscles and adipose tissues [52]. In the myocytes and adipose tissue in mice, improved glucose homeostasis after the ablation of FABP4 was documented [53].

Increased FABP4 production could contribute to macrophage activity, possibly by activation of inflammatory pathways, resulting in inflammation [42]. An alternative hypothesis, more in line with the published evidence, is that the elevated circulating FABP4 reflects the increased cellular production in both adipocytes and macrophages in response to greater lipid availability, with increased Kupffer cell production of FABP4 triggering an increased inflammatory response [42,54].

FABP4 plays a crucial role in mediating the endoplasmic reticulum stress observed in macrophages upon lipotoxic signal exposure, which contributes to atherosclerosis, inflammation, and perhaps plaque vulnerability [55]. High levels of FABP4 may contribute to adverse prognosis via the induction of endoplasmic reticulum stress in macrophages and upregulation of pro-inflammatory cytokine production. Indeed, FABP4 modulates inflammatory responses in macrophages through a positive feedback loop involving c-Jun NH2-terminal kinases and activator protein-1 [42,56]. FABP4 is also implicated in modulating the eicosanoid balance by affecting both cyclooxygenase 2 (COX2) activity and leukotriene A4 (LTA4) stability, and upregulates uncoupling protein 2 (UCP2); all these processes influence macrophage function and adipose tissue inflammation. FABP4 can also interact with HSL and Janus kinase 2 (JAK2) [57] (Figure 1). Two different mechanisms of the latter action have been proposed. The first one includes direct protein–protein interactions between FABP4 and HSL and JAK2 [58,59]. FABP4 and HSL may also interact indirectly via two protein kinases A and G pathways [29].

Several proinflammatory stimuli have been reported to upregulate FABP4 expression in macrophages, including oxidized low-density lipoproteins, toll-like receptor (TLR) agonists, and PPARγ agonists [60,61,62,63]. In this respect, lipopolysaccharide (LPS), a TLR4 ligand, stimulates FABP4 expression and the activated FABP4, reciprocally, enhances the LPS-TLR4 signaling-evoked JNK inflammatory pathways [56]. In addition, prolonged hyperglycemia has been shown to induce FABP4 expression in mesangial cells and trigger the release of proinflammatory cytokines [60,64] (Figure 2).

## 6. Relationship between FABP4 and Insulin Resistance

FABP4 is a cytoplasmic fatty acid chaperone clearly engaged in the onset of insulin resistance [35]. Studies in animal models suggest that FABP4 is important for glucose homeostasis [50]. Deletion of the FABP4 gene protected mice against insulin resistance as well as hyperinsulinemia associated with both diet-induced obesity and genetic obesity [45,50,53,54,65,66]. A reduced ability of adipocytes to take up and retain free fatty acids, leading to ectopic lipid accumulation, and abnormalities in the release of adipokines by adipocytes are critical factors for insulin resistance and the development of T2DM [66].

FABP4 was also detected in apoptotic granulosa cells in atretic antral follicles of the mouse ovary, which suggests a potential link to polycystic ovary syndrome (PCOS), which is known to be frequently associated with insulin resistance [11]. Expression of FABP4 mRNA in isolated granulosa cells was found to be higher in patients diagnosed with PCOS than in controls [67].

It has been proven that FABP4 is negatively correlated with the glucose-disposal rate (GDR) [68]. In non-DM subjects, serum FABP concentrations were negatively correlated with the mean rate of glucose infusion during the last 30 min of the clamp test, which reflects insulin sensitivity [69]. Nakamura et al. [35] also showed that circulating FABP4 concentrations were negatively correlated with GDR, which is a marker of insulin resistance in skeletal muscles in individuals with T2DM. On the contrary, FABP4 concentration was positively related to the insulinogenic index in non-diabetic participants. Wu et al. [70] reported that circulating FABP4 concentrations were correlated with glucose-stimulated insulin secretion in healthy controls.

It has been suggested that insulinotropic potential of FABP4 is similar to the effects of glucagon-like peptide-1 (GLP-1) [35]. To maintain glucose homeostasis, FABP4 may stimulate β cells and alter insulin secretion. Moreover, FABP4 presented a positive relationship with insulin secretion at an early stage in the non-diabetic group, which may be due to the fact that insulin secretion is damaged relatively early in T2DM.

Nakamura et al. [35] found the most pronounced negative correlation between FABP4 and GDR when compared to some other markers of insulin resistance or body composition in T2DM. FABP4 has been reported to have a negative correlation with GDR in type 1 diabetes mellitus, T2DM, and the controls of Asian Americans [71]. FABP4 represents an important molecule dealing with insulin resistance in T2DM.

## 7. Relationship between FABP4 and Gestational Diabetes Mellitus

FABP4 and leptin are known to be involved in the pathophysiology of GDM and its long-term post-partum complications. Placental and non-placental origins of these adipokines are likely to contribute to insulin resistance and β-cell dysfunction [72].

Previous studies found that the serum FABP4 concentrations were significantly increased in women diagnosed with GDM when compared to the control group [65,73,74]. Zhang et al. [65] found that the fasting insulin and age adjusted FABP4 concentrations were significantly higher in the GDM group compared with the normal glucose tolerance participants in the mid and late stages of pregnancy. Furthermore, there was a significant increase in FABP4 from the second to third trimester in women with GDM [65]. Maternal FABP4 concentrations were obviously upregulated in the first trimester in women who later developed GDM [50]. Also, Li et al. [30] observed that FABP4 levels in the GDM group were significantly higher than those of controls, and noted that FABP4 was an independent risk factor for increased insulin resistance during pregnancy. Nevertheless, Ortega-Senovilla et al. [74] did not find any differences in FABP4 levels between the GDM and normal glucose tolerance groups when the FABP4 values were corrected with insulin. Meanwhile, the cited authors revealed significant correlations between maternal blood FABP4 levels and pre-pregnancy BMI values in both control and GDM pregnant patients [74].

Li et al. [32] explained the high circulating FABP4 levels in the maternal serum of pregnant GDM women by its additional release from placenta and adipocytes. Expression of FABP4 mRNA in the placenta and decidua of pregnant women with GDM is greater than that in normal organs [32]. Circulating FABP4 is associated with lipolysis and may aggravate insulin resistance compared to normal physiological insulin resistance during pregnancy [54].

Moreover, candidates for the placental hormones to induce FABP4 overexpression in the placenta and decidua in GDM include human placental lactogen and progesterone [32] as well as the combination of estrogen and progesterone [75]. Their concentrations increase continuously until term and may be associated with an increased insulin resistance along with the progressing gestational age [43,48]. Elevated placental hormones present in serum in GDM may increase the expression of FABP4 mRNA in adipocytes. The synergistic effects of FABP4 from the placenta and adipocytes can affect both metabolic and inflammatory pathways via adipocytes. These actions may play crucial roles in the development of insulin resistance and T2DM in the future lives of post-partum women [32].

On the other hand, FABP4 as expressed in human placental trophoblasts is a key regulator of trophoblastic lipid transport and accumulation during placental development. It has also been reported that maternal FABP4 levels are elevated in preeclampsia, even before the clinical onset of the disease [11]. The level of the second trimester plasma FABP4 in the preeclampsia GDM group was significantly higher than that of the patients with only GDM [76]. Besides, Wotherspoon et al. [77] revealed that increased second-trimester FABP4 levels independently predicted pre-eclampsia in women with type 1 diabetes mellitus.

The serum FABP4 levels were also associated with overweight in GDM patients. Ning et al. [11] concluded that serum FABP4 may be a potential biomarker in GDM diagnosis and is associated with overweight, insulin resistance, and TNF-α in GDM patients.

Our previous study showed that the serum FABP4 levels were significantly higher in the GDM group in the early puerperium in comparison with healthy mothers and women with excessive gestational weight gain [2]. Based on our findings and previous studies, it appears that increased circulating FABP4 concentrations can persist in GDM patients after delivery and might contribute to the increased risk of T2DM and metabolic syndrome. On the other hand, evaluation of FABP4 may be used as a predictive marker for mothers with a history of GDM [2].

In a follow-up study of women six years after GDM, Svensson et al. [78] identified three factors—all related to the adipose tissue and body composition—that, independently of body mass index (BMI) and ethnicity, may increase the risk of progression to T2DM following GDM. These factors included increased serum FABP4 levels, weight gain following index pregnancy, and a lower proportion of fat-free mass. High BMI and abdominal fat distribution were also associated with the development of T2DM after GDM [78].

## 8. Relationship between FABP4 and T2DM and its Complications

A 10-year prospective study proved that high levels of FABP4 at baseline independently predicted the development of T2DM [33]. Serum FABP4 concentrations have been reported to be associated with inadequate glucose control in T2DM [33]. Increased FABP4 levels are also linked to the early presence of metabolic syndrome components, as well as inflammation and oxidation markers in T2DM subjects [44]. Inflammation and oxidative stress are suggested to play a role in the pathogenesis of complications of T2DM [10,79,80].

A previous study showed that the concentrations of FABP4 were negatively associated with endothelial function in T2DM [81]. Another study identified a correlation between FABP4 and impaired endothelial function in diabetes, which lead to an increased cardiovascular risk [82]. Furthermore, the prognostic value of FABP4 for kidney damage [83] and cardiac contractile dysfunction [84] in patients with T2DM have also been proposed. In addition, previous studies suggest that increased FABP4 synthesis in atherosclerotic plaques is associated with disease severity [85,86]. Holm et al. [87] reported that FABP4 was linked to atherogenesis, plaque instability, and adverse outcomes in patients with carotid atherosclerosis and acute ischemic stroke. Two other studies have shown the association of enhanced FABP4 expression within human carotid atherosclerotic lesions with poor prognosis [85,86]. There is available data to suggest that higher levels of FABP4 are also associated with elevated cardiovascular diseases mortality among men with T2DM [42,88].

## 9. Relationship between FABP4 and Diabetic Retinopathy

FABP4 shows clinically relevant potential as a novel predictor of diabetic retinopathy (DR). According to some authors, strict glycemic control and more frequent retinal examination should be recommended for the T2DM patients with the detected highest quartile range of FABP4 [10].

The impact of FABP4 on atherosclerosis seems to be attributed to its action in macrophages [89]. Circulating FABP4 induces insulin resistance, which is an independent biomarker of proliferative retinopathy [90]. Lipopolysaccharides (LPS) stimulate FABP4 transcription through JNK, which in turn induces c-Jun recruitment to a highly conserved activator protein-1 recognition site within the proximal region of the FABP4 promoter [56]. LPS-binding protein is involved in the immune response triggered by inflammatory injury characteristic of DR. Moreover, FABP4 is an obligatory mediator coupling toxic lipids (i.e., saturated fatty acids) to endoplasmic reticulum stress in macrophages in vitro and in vivo [55]. Interestingly, endoplasmic reticulum stress represents an initial event in retina pathogenesis in diabetes [10,91].

## 10. Relationship between FABP4 and Diabetic Nephropathy

In the past decades, several biomarkers have emerged for the detection of early diabetic nephropathy (DN) besides the glomerular filtration rate (GFR) and urine albumin-to-creatinine ratio (UACR). Among them, FABP4 has attracted increased attention [92]. FABP4 was also reported at increased concentrations in nondiabetic as well as T2DM patients with end-stage renal disease [93]. Yeung et al. [94] reported that serum levels of FABP4 had a significantly inverse relationship with the estimated GFR (eGFR) and was independently associated with macrovascular complications and DN staging classified by albuminuria.

Cabré et al. [73] reported that FABP4 was independently associated with eGFR in T2DM patients with eGFR ≥ 60 mL/min/1.73 m^2^. Ni X et al. [92] found that serum FABP4 along with UACR or a panel of biomarkers might be more sensitive for the detection of early DN. Serum FABP4 had an inverse correlation with GFR and could be an independent predictor for early DN [92].

The mechanisms behind the elevation of FABP4 in patients with diabetic kidney disease are not yet fully understood. It is known that FABP4 is abundantly expressed in adipocytes, macrophages, and endothelial cells [92]. Firstly, it is suggested that, during the early stage of DN, the accumulation of active macrophages is more evident in the kidney because of the increased oxidative stress and chronic inflammation, which consequently induces greater expression of serum FABP4 [44,94]. Secondly, damage to glomeruli and tubulointerstitium might result in both decreased glomerular filtration and increased tubular reabsorption, leading to an increase in FABP4 in the circulation [83]. Okazaki et al. [16] reported that urinary excretion of FABP4 was associated with the progression of proteinuria and renal dysfunction in healthy subjects. The cited authors suggested that the urinary FABP4 reflects damage of the glomerular with the hypothesis proposed by Tanaka et al. [13] that the main source of the urinary FABP4 is derived from ectopic expression of glomerular FABP4 rather than increased adiposity and that locally increased FABP4 in the glomerulus affects renal dysfunction.

Toruner et al. [93] found that serum FABP4 levels were independently and positively associated with the albumin excretion rate in patients with T2DM, suggesting an involvement of the increased serum FABP4 levels in the occurrence and development of microalbuminuria among patients with T2DM. Furthermore, researchers from Hong Kong [94] also documented that among patients with diabetes mellitus, serum FABP4 levels were shown to be independently associated with the severity of nephropathy. Their findings raised the possibility that FABP4 might be used as a serum biomarker for stratifying nephropathy stages in patients with T2DM. The serum FABP4 level can be used as an indicator of microalbuminuria not only in diabetic patients with early stage disease, but also for hyperglycemic individuals before the onset of diabetes [95].

## 11. Relationship between FABP4 and Non-Alcoholic Fatty Liver Disease

There are multiple reports available that clearly link metabolic disorders, including insulin resistance and T2DM, to non-alcoholic fatty liver disease (NAFLD), which nowadays represents the most frequent liver disease worldwide. Its incidence has been estimated at 20–30% in the populations of Western countries. Approximately 70% of T2DM and obese subjects present some extent of NAFLD. This abnormal lipid accumulation in the liver is considered a significant causative factor of cancerogenesis resulting in hepatocellular carcinoma (HCC) [96].

At present, HCC represents the third leading cause of cancer-related mortality. Interestingly, except for the well-known risk factors of malignancy, such as viral hepatitis and alcohol, a higher incidence of HCC was observed in T2DM patients. In the analysis, obese subjects and patients with metabolic syndrome were also at a higher risk of developing cancer [96]. Additionally, HCC treatment outcomes of all those patients appeared worse with T2DM, representing a prognostic predictor of the increased risk of mortality [96].

In a study by Thompson et al., the FABP4 expression was indeed proven to be significantly increased in animal models of obesity promoted HCC [97]. These findings are consistent with the previous report, which concluded that PPARγ is considered a tumor suppressor gene, whereas FABP4 plays a role in tumorigenesis [35].

## 12. Conclusions

FABP4, an intracellular lipid chaperone, is highly expressed in adipocytes and macrophages [98]. Abnormalities in the level of FABP4 have been correlated with the development of adiposity, oxidative stress, and atherosclerosis [40]. FABP4 is strongly involved in glucose and lipid metabolism, inflammation, and insulin resistance [76]. In one of the most recent systematic reviews by Bellos et al., targeting the potential association of 10 novel adipokines (i.e., apelin, chemerin, FABP4, fibroblast growth factor-1, monocyte chemoattractant protein-1, nesfatin-1, omentin-1, resistin, vaspin, and visfatin) with GDM, only FABP4 was found to be the most promising predictor of this metabolic pregnancy complication [99].

In fact, a large number of studies have shown that plasma levels of FABP4 are increased in obesity and T2DM, and that circulating FABP4 concentration correlated with clinical outcomes, such as body mass index, insulin resistance, and dyslipidemia [33]. Since the plasma-circulating FABP4 can modulate the function of several types of cells, it becomes extremely interesting to try to develop effective therapeutic strategies against the pathogenesis of metabolic and vascular diseases in this respect [33].

Furthermore, several studies have shown that serum FABP4 levels of GDM patients were higher than those found in women with normal pregnancies [30,65,73,76]. According to the current state of scientific knowledge, females with a previous history of GDM are much more prone to suffering from T2DM, obesity, and metabolic syndrome in the future. There is a huge need to use current research results regarding improved GDM management strategies, including primary prevention for mothers who are at risk of developing subsequent complications. Considering previous findings, it seems that FABP4 may be used as a predictive marker for mothers with a history of GDM. Further studies should focus on the evaluation of FABP4 in the pathogenesis of T2DM following GDM.

## Figures and Tables

**Figure 1 cells-08-00227-f001:**
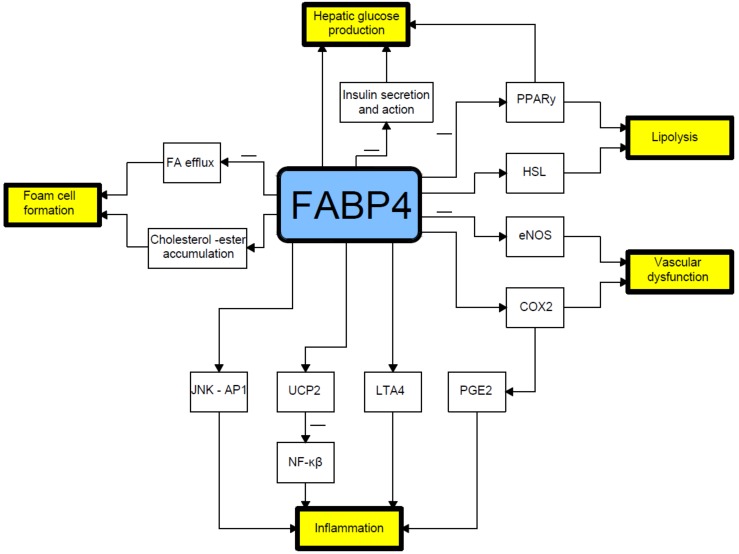
Relationship between fatty acid-binding protein 4 and pathophysiology of type 2 diabetes mellitus. COX2- cyclooxygenase-2; eNOS- endothelial nitric oxide synthase; FA- fatty acid; HSL- hormone sensitive lipase; JNK AP1-Jun N-terminal kinase-activator protein 1; LTA4-leukotriene A4; NF-κB-nuclear factor-kappa B; PGE2-prostaglandin E2; PPARγ–peroxisome proliferator-activated receptor γ; UCP2-uncoupling protein 2.

**Figure 2 cells-08-00227-f002:**
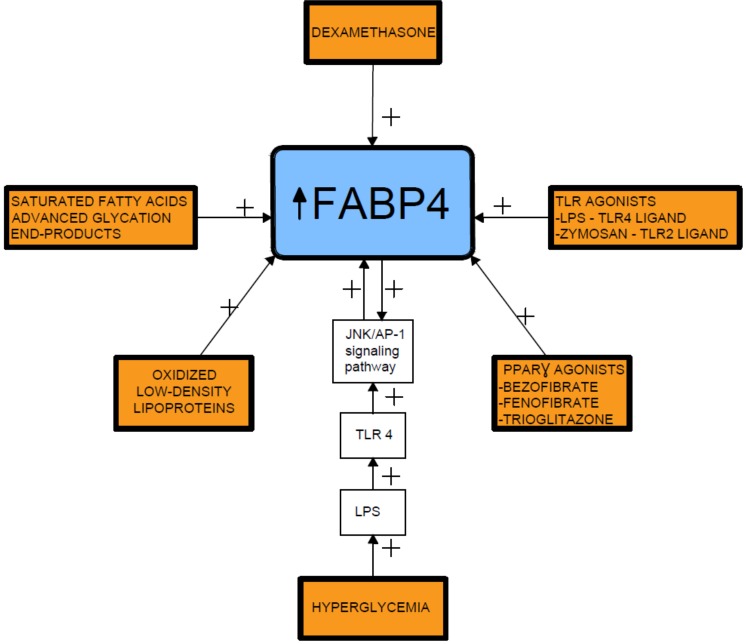
Fatty acid-binding protein 4 inducing factors. TLR- toll-like receptor; LPS- lipopolysaccharide; FABP4-fatty acid-binding protein 4.

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
