# Peer review of "Associations between Fatty Acid-Binding Protein 4–A Proinflammatory Adipokine and Insulin Resistance, Gestational and Type 2 Diabetes Mellitus"

_cells, 2019, doi:10.3390/cells8030227_

Round 1

Reviewer 1 Report

This review presents evidence for the use of FABP4 as a prognostic biomarker and causative factor for obesity, type 2 diabetes and gestational diabetes. The rationale behind the review is strong, though I have a number of comments that I believe will strengthen the manuscript.

Throughout the introductory sections, the link to diabetes needs to be strengthened to make it clear why certain facts have been included - in parts, it just reads as a list of statements rather than a comprehensive argument (e.g. paragraph beginning line 63).

Line 72 - Are CEPB and /or PPARg upregulated in T2DM / GDM?

Line 78 - Insulin downregulates FABP4 levels. Does treatment of diabetic patients with insulin normalise serum levels of FABP4?

Line 93 - Do you mean increase in PPARg expression, instead of FABP4?

Line 101 - 'FAB4' or 'FABP4'?

Line 101 - FABP4 reduces expression of PPARg and adipogenesis, but the PPARg level is reduced in obese patients who presumably have more adipocytes - how do you reconcile these facts?

Line 107 - What does tumorigenesis have to do with diabetes or metabolic syndrome?

Line 163 - FABP4 interacts with HSL and JAK2, but what does this do?

Line 168 - The figure is a useful summary, but is presented in a very basic way. The aesthetics could be improved.

Line 169 - The information in this paragraph on factors that can induce FABP4 would also benefit from being represented as a figure

Line 198 - This suggests FABP4 can increase insulin, and on page 2 the ability of insulin to reduce FABP4 is presented, suggesting this is a negative feedback loop. What causes the disruption to this in diabetes?

Line 232 - How do human placental lactogen and progesterone increase FABP4 expression? Are there response elements to these hormones in the FABP4 promoter, or is the increase in mRNA expression mediated via a different transcription factor?

Line 239 - Is there any difference in the expression levels of FABP4 in non-diabetic preeclampsia patients, and patients with GDM?

Line 277 - This section seems to have a lot of commonalities between cardiovascular disease and diabetic retinopathy. Are there any factors specific to retinopathy? If not, it may be better to combine the two sections.

Intensive glucose control can ameliorate the microvascular complications of diabetes (retinopathy, nephropathy) but macrovascular complications such as atherosclerosis are more resistant. How do circulating FABP4 levels in tightly controlled diabetic patients compare to those who are newly diagnosed or are poorly controlled? 

Minor - replace 'dependant' with 'dependent' throughout the article. JNK is defined on multiple occasions.

Author Response

 Dear Sir, 

We would like to express our gratitude for your meaningful and helpful comments which have made a substantial contribution to the quality of our paper.

Having studied the Reviewers’ comments and following your advice, we have decided to introduce some changes into our paper, which, we do hope, will bring some improvement to our manuscript.

Our responses to the Reviewer's comments:

This review presents evidence for the use of FABP4 as a prognostic biomarker and causative factor for obesity, type 2 diabetes and gestational diabetes. The rationale behind the review is strong, though I have a number of comments that I believe will strengthen the manuscript.

Throughout the introductory sections, the link to diabetes needs to be strengthened to make it clear why certain facts have been included - in parts, it just reads as a list of statements rather than a comprehensive argument (e.g. paragraph beginning line 63).

We have modified the introductory section, so that it corelates with previous paragraphs.

“The human FABP4 consists of 132 amino acids. Its molecular mass has been assessed at 14.6 kDa. FABP4 expression markedly increases at the time of adipocyte differentiation [12]. Due to the abovementioned observation this molecule has been suggested an adipocyte differentiation marker [19]. FABP4 expression is also enhanced during differentiation from monocytes to macrophages. A wide spectrum of different proinflammatory factors modify and control the expression of FABP4 in these cells [20]. In macrophages FABP4 stimulates the foam cell formation. Foam cell formation, which is believed to be mediated by modified low density lipoproteins (LDLs), often occurs in the presence of increased concentrations of insulin and glucose. These increased concentrations are characteristic of insulin resistance associated with diabetes, obesity and the metabolic syndrome [21]. Siersbaek et al. [22] found that peroxisome proliferator-activated receptor γ (PPARγ) and CCAAT/enhancer-binding protein (C/EBP) regulate the expression of most genes associated with adipogenesis. PPARγ not only promotes the proliferation and differentiation of adipocytes but also confers insulin sensitivity to adipocytes [23]. An increase in insulin sensitivity can in turn promote the expression of the PPARγ gene in the adipose tissue, thereby positively accelerating the differentiation of adipocytes. FABP4 expression is controlled at the transcriptional level by PPARγ and C/EBP [12]. However the aforementioned transcriptional regulators seem to be dysregulated in T2DM [24,25].

Other actions encompass modification of the inflammatory response mediated by activation of the IKK-NF-κB and c-Jun N-terminal kinase (JNK)-activator protein-1 (AP-1) pathways [20]. In addition, FABP4 enhances the hydrolytic activity of hormone-sensitive lipase (HSL) [26].

We have also added five references:

Shashkin, P.N.; Jain, N.; Miller, Y.I.; Rissing, B.A.; Huo, Y.; Keller, S.R.; Vandenhoff, G.E.; Nadler, J.L.; McIntyre, T.M. Insulin and glucose play a role in foam cell formation and function. Cardiovasc. Diabetol. 2006, 5, 13.

Siersbaek, R.; Nielsen, R.; Mandrup, S. PPARgamma in adipocyte differentiation and metabolism—novel insights from genome-wide studies. FEBS Lett. 2010, 584, 3242-3249. doi: 10.1016/j.febslet.2010.06.010.

Huang, Q.; Ma, C.; Chen, L.; Luo, D.; Chen, R.; Liang, F. Mechanistic Insights Into the Interaction Between Transcription Factors and Epigenetic Modifications and the Contribution to the Development of Obesity. Front. Endocrinol. (Lausanne). 2018, 9, 370. doi: 10.3389/fendo.2018.00370.

Mizukami, H.; Takahashi, K.; Inaba, W.; Tsuboi, K.; Osonoi, S.; Yoshida, T.; Yagihashi, S. Involvement of oxidative stress-induced DNA damage, endoplasmic reticulum stress, and autophagy deficits in the decline of β-cell mass in Japanese type 2 diabetic patients. Diabetes Care 2014, 37, 1966-1974.

Lehrke, M.; Lazar, M.A. The many faces of PPARg. Cell 2005, 123, 993–999.

Line 72 - Are CEPB and /or PPARg upregulated in T2DM / GDM?

PPARγ is a critical transcriptional regulator of adipogenesis in mammals, closely related to regulation of lipids and glucose metabolism The development of obesity, T2DM and atherosclerosis is associated with dysregulation of PPARγ [Lehrke and Lazar, 2005].

Mizukami et al. [2014] observed the accumulation of C/EBPβ in the pancreatic β cells of patients with T2DM.

We have added the piece of information to the statement:

“FABP4 expression is controlled at the transcriptional level by PPARγ and C/EBP [12]. However the aforementioned transcriptional regulators seem to be dysregulated in T2DM [24,25].

Line 78 - Insulin downregulates FABP4 levels. Does treatment of diabetic patients with insulin normalise serum levels of FABP4?

We have changed this part of the manuscript:

“FABP4 as an adipokine may influence insulin sensitivity. On the other hand, expression of FABP4 is highly induced during adipocyte differentiation and transcriptionally controlled by PPARγ agonists, fatty acids, dexamethasone, and insulin [27]. Insulin downregulates only microvesicle-free-mediated and microvesicle-secreted FABP4. However, the release of FABP4 via adipocyte-derived microvesicles is a small fraction and conveys a minor activity [27].

Furthermore, FABP4-/- animals exhibit a defect in β-adrenergic stimulated insulin secretion, even under lean conditions, suggesting an effect on β cell function, another cell type that does not express FABP4. This is complemented by human studies showing that higher serum FABP4 levels correlate with higher insulin response index in T2DM patients, and a higher insulinogenic index in non-diabetics [28]. In vivo, FABP4 levels are suppressed under refeeding or insulin [28].

We have also added two new references:

Furuhashi, M. Fatty Acid-Binding Protein 4 in Cardiovascular and Metabolic Diseases. J. Atheroscler. Thromb. 2019. doi: 10.5551/jat.48710.

Prentice, K.J.; Saksi, J.; Hotamisligil, G.S. Adipokine FABP4 integrates energy stores and counter regulatory metabolic responses. J. Lipid Res. 2019. pii: jlr.S091793. doi: 10.1194/jlr.S091793.

Line 93 - Do you mean increase in PPARg expression, instead of FABP4?

We have modified the sentence. Indeed, we meant PPARg (the spelling error).

Line 101 - 'FAB4' or 'FABP4'?

We have corrected the indicated word.

Line 101 - FABP4 reduces expression of PPARg and adipogenesis, but the PPARg level is reduced in obese patients who presumably have more adipocytes - how do you reconcile these facts?

As we mentioned in our paper - a large number of studies have shown that plasma levels of FABP4 are increased in obesity and T2DM, and that circulating FABP4 concentration correlated with clinical outcomes, such as body mass index, insulin resistance and dyslipidemia [26 33]. According to the Reviewer's comment, FABP4 reduces expression of PPARγ and adipogenesis. The FABP4 level was higher and PPARγ level was lower in human visceral fat and mouse epididymal fat compared with their subcutaneous fat. Furthermore, FABP4 was higher in the adipose tissues of obese diabetic individuals compared with healthy ones [28 35].

Line 107 - What does tumorigenesis have to do with diabetes or metabolic syndrome?

According to the suggestion of the Reviewer 2 the section entitled: “11. Relationship between FABP4 and non-alcoholic fatty liver disease” was added. The abovementioned sentence was included in this paragraph.

There are multiple reports available that clearly link metabolic disorders, including insulin resistance and T2DM, to non-alcoholic fatty liver disease (NAFLD), which nowadays represents the most frequent liver disease worldwide. Its incidence has been estimated at 20-30% in populations of the Western countries. Approximately 70% of T2DM and obese subjects present some extent of NAFLD. This abnormal lipid accumulation in liver is considered a significant causative factor of cancerogenesis resulting in hepatocellular carcinoma (HCC) [96].

At present, HCC represents the third leading cause of cancer-related mortality. Interestingly, except for the well-known risk factors of malignancy such as viral hepatitis and alcohol, the higher incidence of HCC was observed in T2DM patients. In the analysis the obese subjects and patients with metabolic syndrome were also at a higher risk of developing cancer. Additionally, HCC treatment outcome of all those patients appeared worse with T2DM representing a prognostic predictor of the increased risk of mortality [96].

In a study by Thompson et al. the FABP4 expression was indeed proved to be significantly increased in animal models of obesity promoted HCC [97]. These findings are consistent with the previous report which concluded that PPARγ is considered a tumor suppressor gene, whereas FABP4 plays a role in tumorigenesis [35].”

We have also changed the order of references.

Line 163 - FABP4 interacts with HSL and JAK2, but what does this do?

Two different mechanisms of FABP4 interactions with HSL have been proposed. The first one acompasses two protein kinases A and G pathways. The second mechanism is known to be a direct protein–protein interaction between FABP4 and HSL, in which a domain of FABP4 and an HSL binding site of the activated, phosphorylated HSL interact [Smith et al., 2007; Furuhashi et al., 2015].

In case of JAK2, FABP4 has been reported to signal via protein–protein interaction [Thompson et al., 2009].

The mechanisms of FABP4 interactions with HSL and JAK2 have been detailed in the text as follows:

Two different mechanisms of the latter action have been proposed. The first one includes direct protein–protein interactions between FABP4 and HSL and JAK2 [58,59]. FABP4 and HSL may also interact indirectly via two protein kinases A and G pathways [22 29].

We have also added two references:

Smith, A.J.; Thompson, B.R.; Sanders, M.A.; Bernlohr, D.A. Interaction of the adipocyte fatty acid-binding protein with the hormone-sensitive lipase: regulation by fatty acids and phosphorylation. J. Biol. Chem. 2007, 282, 32424-32432.

Thompson, B.R.; Mazurkiewicz-Muñoz, A.M.; Suttles, J.; Carter-Su, C.; Bernlohr, D.A. Interaction of adipocyte fatty acid-binding protein (AFABP) and JAK2: AFABP/aP2 as a regulator of JAK2 signaling. J. Biol. Chem. 2009, 284, 13473-13480. doi: 10.1074/jbc.M900075200.

Line 168 - The figure is a useful summary, but is presented in a very basic way. The aesthetics could be improved.

 The Figure 1 was modified as requested.                                               

 Line 169 - The information in this paragraph on factors that can induce FABP4 would also benefit from being represented as a figure.

 The Figure 2 entitled “Fatty acid-binding protein 4 inducing factors” was added.

Line 198 - This suggests FABP4 can increase insulin, and on page 2 the ability of insulin to reduce FABP4 is presented, suggesting this is a negative feedback loop. What causes the disruption to this in diabetes?

The available scientific evidence suggests that the abovementioned interactions are different in the healthy and diabetic subjects.

Line 232 - How do human placental lactogen and progesterone increase FABP4 expression? Are there response elements to these hormones in the FABP4 promoter, or is the increase in mRNA expression mediated via a different transcription factor?

In the cited manuscript by Li et al. the suggested mechanism of the increased FABP4 expression is represented by mRNA synthesis stimulation triggered by hPL and progesterone, which was investigated and documented in animal model - embryos in mice. The mentioned authors conclude that elevated placental hormones originating from serum in GDM may increase the expression of FABP4 mRNA in adipocytes. The synergistic effects of FABP4 from the placenta and adipocytes can lead to the development of insulin resistance and type 2 diabetes mellitus. In the light of the aforementioned study it has been proposed that FABP4 plays a role in the establishment and maintenance of pregnancy.

We have also found a study by Wang et al. [2017], in the light of which FABP4 expression is regulated by the combination of estrogen and progesterone, however, it is unknown if FABP4 in human epithelial cell regulates stromal decidualization via paracrine signaling.

We have added this piece of information and the cited reference to our manuscript.

“Moreover, candidates for the placental hormones to induce FABP4 overexpression in the placenta and decidua in GDM include human placental lactogen and progesterone [25 32] as well as the combination of estrogen and progesterone [75].”

Wang, P.; Zhu, Q.; Peng, H.; Du, M.; Dong, M.; Wang, H. Fatty Acid-Binding Protein 4 in Endometrial Epithelium Is Involved in Embryonic Implantation. Cell. Physiol. Biochem. 2017, 41, 501-509. doi: 10.1159/000456886.

Line 239 - Is there any difference in the expression levels of FABP4 in non-diabetic preeclampsia patients, and patients with GDM?

Yan et al. [2016] found that the expression of FABP4 was increased in placentas from patients with preeclampsia when compared to healthy pregnant patients. Maternal serum FABP4 concentrations were significantly elevated in preeclampsia when compared to healthy controls of similar gestational age [Fasshauer et al., 2008].

Li et al. [2018] observed that the level of the second trimester plasma FABP4 in the preeclampsia GDM group was significantly higher than that of the patients with only GDM. Besides, Wotherspoon et al. [2016] revealed that increased second-trimester FABP4 levels independently predicted pre-eclampsia in women with type 1 diabetes mellitus.

We have added this piece of information to our paper:

The level of the second trimester plasma FABP4 in the preeclampsia GDM group was significantly higher than that of the patients with only GDM [76]. Besides, Wotherspoon et al. [77] revealed that increased second-trimester FABP4 levels independently predicted pre-eclampsia in women with type 1 diabetes mellitus.

Yan YPeng HWang PWang HDong M. Increased expression of fatty acid binding protein 4 in preeclamptic Placenta and its relevance to preeclampsia. Placenta. 2016;39:94-100. doi: 10.1016/j.placenta.2016.01.014.

Li, B.; Yang HZhang WShi YQin SWei YHe YYang WJiang SJin H. Fatty acid-binding protein 4 predicts gestational hypertension and preeclampsia in women with gestational diabetes mellitus. PLoS One. 2018, 13, e0192347. doi: 10.1371/journal.pone.0192347.

Fasshauer MSeeger JWaldeyer TSchrey SEbert TKratzsch JLössner UBlüher MStumvoll MFaber RStepan H. Serum levels of the adipokine adipocyte fatty acid-binding protein are increased in preeclampsia. Am J Hypertens. 2008, 21, 582-586. doi: 10.1038/ajh.2008.23.

Wotherspoon, A.C.; Young, I.S.; McCance, D.R.; Patterson, C.C.; Maresh, M.J.; Pearson, D.W.; Walker, J.D.; Holmes, V.A.; Diabetes and Pre-eclampsia Intervention Trial (DAPIT) Study Group. Serum Fatty Acid Binding Protein 4 (FABP4) Predicts Pre-eclampsia in Women With Type 1 Diabetes. Diabetes Care 2016, 39, 1827-1829. doi: 10.2337/dc16-0803.

Line 277 - This section seems to have a lot of commonalities between cardiovascular disease and diabetic retinopathy. Are there any factors specific to retinopathy? If not, it may be better to combine the two sections.

Intensive glucose control can ameliorate the microvascular complications of diabetes (retinopathy, nephropathy) but macrovascular complications such as atherosclerosis are more resistant. How do circulating FABP4 levels in tightly controlled diabetic patients compare to those who are newly diagnosed or are poorly controlled?

If the Reviewer would not mind, we would rather not combine these sections and leave the layout of the manuscript as it is. We believe that the current version helps to enhance the clarity of the paper.

Taking into consideration the fact that different criteria for diabetes diagnosis and assessment of its severity were used by various authors in non-homogenous populations (age, race, etc.), it seems impossible at this stage  to corelate the levels of circulating FABP4 with the degree of clinical control of diabetes. Moreover, it is necessary to stress that FABP4 represents a relatively novel adipokine.

An interesting study by Jahansouz et al. was published in 2018, in which the authors documented the decrease in FABP4 levels following Roux en Y gastric bypass surgery, however intensive life style modification or pharmacotherapy did not have any effect in this respect in patients diagnosed with T2DM.

Jahansouz CXu HKizy S, Thomas AJJosephrajan AHertzel AVFoncea RConnett JCBillington CJJensen MKorner JBernlohr DAIkramuddin S. Serum FABP4 concentrations decrease after Roux-en-Y gastric bypass but not after intensive medical management. Surgery 2018. pii: S0039-6060(18)30517-8. doi: 10.1016/j.surg.2018.08.007.

Minor - replace 'dependant' with 'dependent' throughout the article. JNK is defined on multiple occasions.

We have corrected the indicated word. We have also reduced JNK definition to ones.

We appreciate your time and look forward to your response.

Yours faithfully,

Marcin Trojnar and co-authors

Reviewer 2 Report

In the last years, adipose tissue inflammation has been shown to be one of the major mechanisms underlying adipose tissue dysfunction, contributing to the development of metabolic derangements in other organs, mainly liver.

For this reason, authors should deepen in their review the following aspects:

Excess adiposity favors the development of  type-2 diabetes mellitus, cardiovascular disease, dyslipidemia, NAFLD/NASH, and cancer, i.e., HCC. 

Insulin resistance is a central mechanism to NAFLD/NASH.

Now, being NAFLD one of the main causes of HCC, as evident in...World J Gastroenterol. 2014 Jul 28;20(28):9217-28. Could metabolic syndrome lead to hepatocarcinoma via non-alcoholic fatty liver disease?...this point should be stressed.

To reinforce the above-mentioned issue there is an altered fatty acid-binding protein 4 (FABP4) expression and function in human and animal models of hepatocellular carcinoma, as shown in..... Thompson KJ, et al. Liver Int. 2018.;38: 1074-83

Results suggest that metformin reduces lipid accumulation in macrophages by repressing FOXO1-mediated FABP4 transcription. Thus, metformin may have a protective effect against lipid accumulation in macrophages and may serve as a therapeutic agent for preventing and treating atherosclerosis in metabolic syndrome.

Biochemical and Biophysical Research Communication. 2010; 393(1):89-94

A small molecular ligand for FABP4 that blocks the binding of endogenous ligands may be developed into a drug for the treatment of type-2 diabetes.

Author Response

Dear Sir,

We would like to express our gratitude for your meaningful and helpful comments which have made a substantial contribution to the quality of our paper.

Having studied the Reviewers’ comments and following your advice, we have decided to introduce some changes into our paper, which, we do hope, will bring some improvement to our manuscript.

Our responses to the Reviewer's comments:

In the last years, adipose tissue inflammation has been shown to be one of the major mechanisms underlying adipose tissue dysfunction, contributing to the development of metabolic derangements in other organs, mainly liver.

For this reason, authors should deepen in their review the following aspects:

Excess adiposity favors the development of type-2 diabetes mellitus, cardiovascular disease, dyslipidemia, NAFLD/NASH, and cancer, i.e., HCC.

Insulin resistance is a central mechanism to NAFLD/NASH.

Now, being NAFLD one of the main causes of HCC, as evident in...World J Gastroenterol. 2014 Jul 28;20(28):9217-28. Could metabolic syndrome lead to hepatocarcinoma via non-alcoholic fatty liver disease?...this point should be stressed.

To reinforce the above-mentioned issue there is an altered fatty acid-binding protein 4 (FABP4) expression and function in human and animal models of hepatocellular carcinoma, as shown in.....Thompson KJ, et al. Liver Int. 2018.;38: 1074-83.

We have added the section entitled: “11. Relationship between FABP4 and non-alcoholic fatty liver disease” as well as the following sentences to this section:

“There are multiple reports available that clearly link metabolic disorders, including insulin resistance and T2DM, to non-alcoholic fatty liver disease (NAFLD), which nowadays represents the most frequent liver disease worldwide. Its incidence has been estimated at 20-30% in populations of the Western countries. Approximately 70% of T2DM and obese subjects present some extent of NAFLD. This abnormal lipid accumulation in liver is considered a significant causative factor of cancerogenesis resulting in hepatocellular carcinoma (HCC) [96].

At present, HCC represents the third leading cause of cancer-related mortality. Interestingly, except for the well-known risk factors of malignancy such as viral hepatitis and alcohol, the higher incidence of HCC was observed in T2DM patients. In the analysis the obese subjects and patients with metabolic syndrome were also at a higher risk of developing cancer. Additionally, HCC treatment outcome of all those patients appeared worse with T2DM representing a prognostic predictor of the increased risk of mortality [96].

In a study by Thompson et al. the FABP4 expression was indeed proved to be significantly increased in animal models of obesity promoted HCC [97]. These findings are consistent with the previous report which concluded that PPARγ is considered a tumor suppressor gene, whereas FABP4 plays a role in tumorigenesis [28 35].”

The part of the last sentence was transferred from the section entitled “3. Relationship between FABP4 and PPARγ”.

We have also changed the order of references.

Scalera, A.; Tarantino, G. Could metabolic syndrome lead to hepatocarcinoma via non-alcoholic fatty liver disease? World J. Gastroenterol. 2014, 20, 9217-9228. doi: 10.3748/wjg.v20.i28.9217.

Thompson, K.J.; Austin, R.G.; Nazari, S.S.; Gersin, K.S.; Iannitti, D.A.; McKillop, I.H. Altered fatty acid-binding protein 4 (FABP4) expression and function in human and animal models of hepatocellular carcinoma. Liver Int. 2018, 38, 1074-1083. doi: 10.1111/liv.13639.

Results suggest that metformin reduces lipid accumulation in macrophages by repressing FOXO1-mediated FABP4 transcription. Thus, metformin may have a protective effect against lipid accumulation in macrophages and may serve as a therapeutic agent for preventing and treating atherosclerosis in metabolic syndrome.

Biochemical and Biophysical Research Communication. 2010; 393(1):89-94

We have added the following part to the manuscript:

“It has also been postulated that the transcription factor forkhead box protein O1 (FOXO1) is involved in lipid metabolism. Available scientific evidence suggests that metformin may have a protective effect against lipid accumulation in macrophages as it decreases FABP4 expression at the mRNA level. The exact mechanism is not yet fully understood, it may deal with decrease in transcription or by promotion of mRNA degradation. For this reason some authors suggest that metformin should also be perceived as a therapeutic agent for preventing and targeting atherosclerosis in metabolic syndrome [49].”

We have also changed the order of references:

Song, J.; Ren, P.; Zhang, L.; Wang, X.L.; Chen, L.; Shen, Y.H. Metformin reduces lipid accumulation in macrophages by inhibiting FOXO1-mediated transcription of fatty acid-binding protein 4. Biochem. Biophys. Res. Commun. 2010, 393, 89-94. doi: 10.1016/j.bbrc.2010.01.086.

A small molecular ligand for FABP4 that blocks the binding of endogenous ligands may be developed into a drug for the treatment of type-2 diabetes.

We have added the sentence: “A small molecular ligand for FABP4 that blocks the binding of endogenous ligands may be developed into a drug for the treatment of T2DM [46].” to the manuscript.

Wang, Q.; Rizk. S.; Bernard, C.; Lai, M.P.; Kam, D.; Storch, J.; Stark, R.E. Protocols and pitfalls in obtaining fatty acid-binding proteins for biophysical studies of ligand-protein and protein-protein interactions. Biochem. Biophys. Rep. 2017, 10, 318-324. doi: 10.1016/j.bbrep.2017.05.001.

We have also changed the order of references.

We appreciate your time and look forward to your response.

Yours faithfully,

Marcin Trojnar and co-authors

Round 2

Reviewer 1 Report

Thank you for addressing my comments - I am happy for the article to be published in it's present form.

Author Response

Dear Reviewer,

Thank you very much for your kind opinion regarding our manuscript.

We appreciate your time.

Yours sincerely,

Marcin Trojnar, M.D., Ph.D.

the Chair and Department of Internal Medicine

Medical University of Lublin

Staszica 1, 20-081 Lublin, Poland

Tel: +48 5327717, E-mail: [email protected]